# Optimal Positive Generation via Latent Transformation for Contrastive Learning

**Yinqi Li**[1,2], **Hong Chang**[1,2], **Bingpeng Ma**[2], **Shiguang Shan**[1,2], **Xilin Chen**[1,2]
[1]Institute of Computing Technology, Chinese Academy of Sciences
[2]University of Chinese Academy of Sciences
`yinqi.li@vipl.ict.ac.cn`, `{changhong, sgshan, xlchen}@ict.ac.cn`, `bpma@ucas.ac.cn`

## Abstract

Contrastive learning, which learns to contrast positive with negative pairs of samples, has been popular for self-supervised visual representation learning. Although great effort has been made to design proper positive pairs through data augmentation, few works attempt to generate optimal positives for each instance. Inspired by semantic consistency and computational advantage in latent space of pretrained generative models, this paper proposes to learn instance-specific latent transformations to generate Contrastive Optimal Positives (COP-Gen) for self-supervised contrastive learning. Specifically, we formulate COP-Gen as an instance-specific latent space navigator which minimizes the mutual information between the generated positive pair subject to the semantic consistency constraint. Theoretically, the learned latent transformation creates optimal positives for contrastive learning, which removes as much nuisance information as possible while preserving the semantics. Empirically, using generated positives by COP-Gen consistently outperforms other latent transformation methods and even real-image-based methods in self-supervised contrastive learning.

## 1  Introduction

Learning effective visual representation without explicit human annotation is attractive but challenging in machine learning and computer vision. One way to achieve this goal is through self-supervised learning, which leverages handcrafted pretext tasks to create the learning objectives, such as context prediction [1], colorization [2], rotation prediction [3], and generative modeling [4, 5].

Recently, contrastive learning [6, 7, 8, 9, 10, 11] has achieved great success in self-supervised visual representation learning. The goal of contrastive learning is to pull semantically similar samples (positive pairs) together in the feature space while pushing dissimilar samples (negative pairs) apart. Without access to labels, negative pairs are sampled randomly and in large amount [10, 11] to obtain effective representations [12, 13]. A line of works are proposed to calibrate the impact of negative samples [14, 15, 16, 17, 18, 19].

On the other hand, the construction of positive pairs is also very crucial. Tian et al. [20] demonstrates that optimal positive pairs are task-dependent. Since the labels of downstream tasks and pretraining dataset are not accessible in self-supervised learning, a common way to create positive pairs is to use semantic-preserving data space transformation (augmentation) to obtain two or more different views of the same sample, as shown in Figure 1a. Great effort has been made to find the optimal augmentation policy [11, 20]. Nevertheless, data augmentation cannot create **instance-specific optimal** positives, which are key in contrastive learning to preserve semantic and discard nuisance information for each sample. Some works try to address this problem by learning adversarial

---

Code and models are available at: https://github.com/LiYinqi/COP-Gen

36th Conference on Neural Information Processing Systems (NeurIPS 2022).

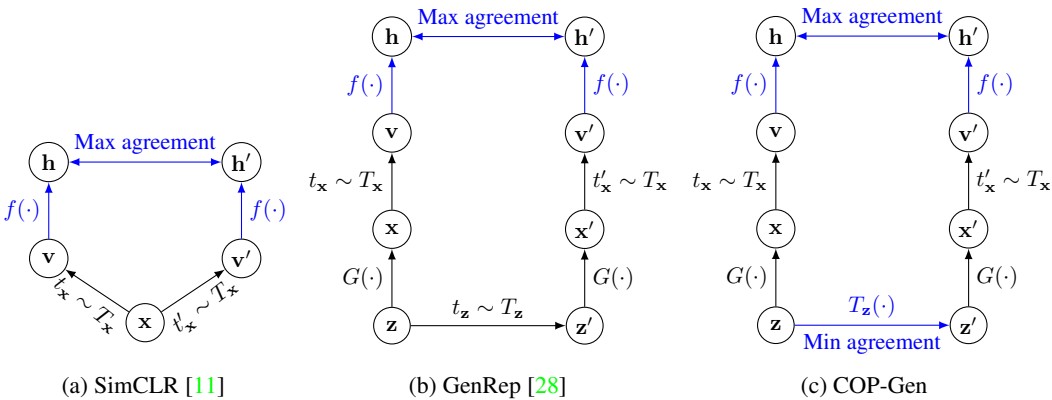

Figure 1: Different ways of creating positive pairs for contrastive learning. (a) is based on data space transformation $T_\mathbf{x}$ while (b) (c) based on latent space transformation $T_\mathbf{z}$. Our proposed COP-Gen aims to create optimal positives for each anchor while (a) (b) do not. Blue arrows indicate learnable and black non-learnable operations. Figure inspired by [11].

perturbations in data space [21, 22, 23] or using nearest neighbors in the learned representation space [24, 25, 26, 27]. However, the former limits the transformation diversity and the latter suffers from a high false positive rate [25].

Instead of creating optimal positives for each instance in data space or representation space, **a more proper way to address this problem is through transformation in the latent space of well-trained generative models.** The main reasons include (1) semantic consistency between the positive pair can be achieved in the latent space (in contrast to representation space transformation and non-domain-knowledge-guided data augmentation) by controlling the transformation distance within a small range; (2) latent space transformation is continuous and differentiable (in contrast to data augmentation), which is computationally friendly for searching instance-wise optimal transformations. In addition, it may bring more diversity beyond data augmentation. Jahanian et al. [28] have made a first step towards leveraging latent space transformation together with data augmentation to create positive pairs, as shown in Figure 1b. However, the transformation is still designed for the whole dataset. How to find the optimal latent transformation for each sample remains an open issue.

In this work, to address the above problems, we propose an instance-specific Contrastive Optimal Positive Generation (COP-Gen) approach via latent space transformation. Specifically, COP-Gen learns a navigator in the latent space, whose objective is to transform each input anchor to the optimal position. To this end, subjecting to the semantic constraint, the navigator is simply learnt to minimize the mutual information between the generated positive pair. Therefore, the training process is implemented by adversarially training a mutual information estimator together with the latent space navigator, as illustrated in Figure 1c. We summarize our contributions as follows:

- We find the latent space of well-trained generative model fits exactly for generating instance-specific optimal positives in fully unsupervised setting and thus propose COP-Gen.

- We theoretically demonstrate that the learned latent transformation generates optimal positive pairs for contrastive learning, which can preserve semantic-relevant information while discarding nuisance information.

- Extensive experiments show that using COP-Gen to generate positives outperforms other latent transformation methods and even real-image-based methods in self-supervised contrastive learning. We also extend COP-Gen to supervised contrastive learning.

## 2 Preliminaries

**Contrastive Learning.** Given a training sample $\mathbf{x}$, state-of-the-art contrastive methods like SimCLR [11], as illustrated in Figure 1a, use data augmentations $t_\mathbf{x} \sim T_\mathbf{x}$ and $t'_\mathbf{x} \sim T_\mathbf{x}$ to create a positive pair $(\mathbf{v}, \mathbf{v}')$, and then learn an encoder $f$ to maximize the agreement between them by contrasting

with $K$ randomly sampled negatives, using the InfoNCE loss [9]:

$$\mathcal{L}_{\text{NCE}}^f(\mathbf{v}, \mathbf{v}') = -\mathbb{E}\left[\log \frac{\exp(\text{sim}(f(\mathbf{v}), f(\mathbf{v}')))}{\exp(\text{sim}(f(\mathbf{v}), f(\mathbf{v}'))) + \sum_{k=1}^{K} \exp(\text{sim}(f(\mathbf{v}), f(\mathbf{v}'_k)))}\right], \quad (1)$$

where $\text{sim}$ denotes the cosine similarity between two vectors. It is proved that minimizing InfoNCE loss maximizes a lower bound on mutual information [9, 29]: $I(\mathbf{v}; \mathbf{v}') \geq \log(K) - \mathcal{L}_{\text{NCE}}^f$, where $I(\mathbf{v}; \mathbf{v}')$ represents mutual information between $\mathbf{v}$ and $\mathbf{v}'$.

**Optimal Positives for Contrastive Learning.** Intuitively, the data augmentation $T_{\mathbf{x}}$ that creates positive pairs should preserve the semantic characteristics while changing the nuisance aspects [30]. To find the right balance, Tian et al. [20] introduce an "InfoMin principle", which demonstrates that optimal positives for contrastive learning are task-dependent.

**Definition 2.1** (Optimal Positives for Contrastive Learning, Proposition A.1 in [20]). Given a downstream task $\mathcal{T}$ with label $\mathbf{y}$, the optimal positive pair $(\mathbf{v}^*, \mathbf{v}'^*)$ created from the data $\mathbf{x}$ satisfies

$$I(\mathbf{v}^*; \mathbf{v}'^*) = I(\mathbf{x}; \mathbf{y}), \quad (2)$$
$$I(\mathbf{v}^*; \mathbf{v}'^* | \mathbf{y}) = 0, \quad (3)$$

subject to $I(\mathbf{v}; \mathbf{y}) = I(\mathbf{v}'; \mathbf{y}) = I(\mathbf{x}; \mathbf{y})$.

Definition 2.1 indicates that when we have access to the downstream task, the optimal positive pair should only capture the task-relevant information $I(\mathbf{x}; \mathbf{y})$, while leaving the nuisance information $I(\mathbf{v}^*; \mathbf{v}'^* | \mathbf{y})$ to be zero.

Unfortunately, $\mathbf{y}$ is not accessible in unsupervised contrastive learning. The construction of positive pairs is commonly guided by domain knowledge or experimental results instead [11, 20] and could not be instance-wise optimal. To address this problem, we will next develop an optimal positive generation approach that uses pretrained generative models without accessing the labels.[1]

## 3 Contrastive Optimal Positive Generation via Latent Transformation

In this section, we first introduce our contrastive optimal positive generation (COP-Gen) approach in Section 3.1. Next, we give a theoretical analysis in Section 3.2. In Section 3.3 we describe and explain the practical details.

### 3.1 Motivation and COP-Gen Approach

Before describing our approach, we explain the inspiration behind it by formulating a prominent property of pretrained generative models. It has been observed that using pretrained generative models can create images with interpretable transformations [31, 32, 33], which has several interesting applications such as face attribute editing [34] and data augmentation [35]. These generative models [31, 36, 37] are commonly trained on annotated or one-class datasets. Yet it is worth noting that even they are trained on unlabeled large scale dataset like ImageNet [38], nearby latent vectors can map to semantically similar images, which is a remarkable property for unsupervised representation learning [28]. See Figure 2 for an empirical observation. We formulate this property as follows.

**Proposition 3.1** (Semantic Consistency in Latent Space). *Given a well-trained unconditional generative model $G$, $\mathbf{z}$ and $\mathbf{z}'$ are two latent vectors, if the distance $d(\mathbf{z}, \mathbf{z}') \leq \delta$, then $G(\mathbf{z})$ and $G(\mathbf{z}')$ will have the similar semantic label $\mathbf{y}$. In the form of mutual information, we have*

$$|I(G(\mathbf{z}); \mathbf{y}) - I(G(\mathbf{z}'); \mathbf{y})| \leq \varepsilon, \quad (4)$$

*where $\varepsilon$ stands for tolerable semantic difference, and $\delta$ is the maximum shifted distance to maintain semantic consistency (approximately corresponding to the second column of $G(\mathbf{z}')$ in Figure 2).*

The above observation and Proposition 3.1 inspire us to find optimal positives in the latent space of well-trained generative models without access to the labels. Therefore, we propose the following

---

[1]The optimal positives for contrastive pretraining always depend on *downstream task*, which is usually unknown in advance. We here focus on addressing the label-inaccessible issue of *pretraining* dataset, which is assumed to be helpful and general.

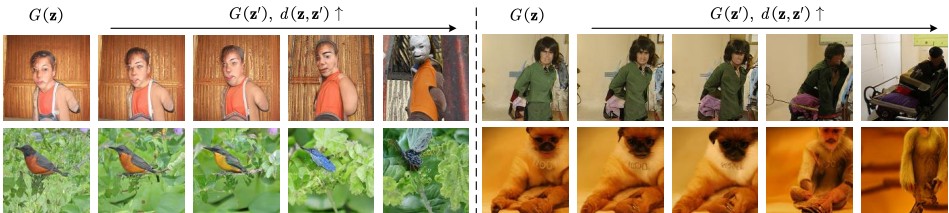

Figure 2: Randomly sampled images from ImageNet-pretrained unconditional BigBiGAN [39]. From left to right for each example, the distance between $\mathbf{z}$ and $\mathbf{z}'$ keeps rising.

COP-Gen approach. Our goal is to learn a latent space navigator $T_\mathbf{z}(\mathbf{z})$, which can map each sampled anchor $\mathbf{z}$ to the right position $\mathbf{z}'$, where the constructed pair $(G(\mathbf{z}), G(\mathbf{z}'))$ is the optimal for contrastive learning.

**COP-Gen.** Our idea, inspired by the InfoMin principle, is to reduce as much shared information between $G(\mathbf{z})$ and $G(T_\mathbf{z}(\mathbf{z}))$ as possible while preserving the semantics:

$$\min_{T_\mathbf{z}} I(t_\mathbf{x}(G(\mathbf{z})); t'_\mathbf{x}(G(T_\mathbf{z}(\mathbf{z})))), \text{ s.t. } d(\mathbf{z}, T_\mathbf{z}(\mathbf{z})) \le \delta, \text{ where } \mathbf{z} \sim p_\mathbf{z}, t_\mathbf{x}, t'_\mathbf{x} \sim T_\mathbf{x}. \quad (5)$$

In practice, a shared encoder $f$ trained with InfoNCE loss is applied to estimate a lower bound of the mutual information between the positive samples, as illustrated in Figure 1c. Specifically, Eq. (5) is implemented as the following adversarial training form:

$$\max_{T_\mathbf{z}} \min_f \mathcal{L}^f_{\text{NCE}}(t_\mathbf{x}(G(\mathbf{z})), t'_\mathbf{x}(G(T_\mathbf{z}(\mathbf{z})))), \text{ s.t. } d(\mathbf{z}, T_\mathbf{z}(\mathbf{z})) \le \delta, \text{ where } \mathbf{z} \sim p_\mathbf{z}, t_\mathbf{x}, t'_\mathbf{x} \sim T_\mathbf{x}. \quad (6)$$

After learning $T_\mathbf{z}$, we freeze it and use it together with $G$ to generate positive pairs offline. After that, we perform the standard contrastive learning by training an encoder that maximizes (a lower bound on) the mutual information between generated positive pairs. As will be proved in Section 3.2, there is no nuisance information remaining in any generated positive pairs to be learned by the encoder.

Note that the synthetic images are usually object-centric and color-biased [40]. As a consequence, when using these synthetic images for contrastive learning, data space augmentations $T_\mathbf{x}$ like random cropping and color jittering cannot be replaced even when the latent transformation is optimal. Thus, $T_\mathbf{x}$ is also adopted in COP-Gen to keep consistency with the contrastive learning stage.

### 3.2 Theoretical Analysis

In this subsection, we theoretically demonstrate that under the guarantee of semantic consistency, COP-Gen will find optimal latent transformations for each sampled latent vector. We first give the following assumption and lemma.

**Assumption 3.2** (Semantic-preserving $T_\mathbf{x}$). The data space augmentation $T_\mathbf{x} : \mathcal{X} \to \mathcal{X}$ should be designed to be semantic-preserving. In the form of mutual information,

$$I(t_\mathbf{x}(\mathbf{x}); \mathbf{y}) = I(\mathbf{x}; \mathbf{y}), \forall t_\mathbf{x} \sim T_\mathbf{x}, \quad (7)$$

where $\mathbf{x} \in \mathcal{X}$ is the data, $\mathbf{y} \in \mathcal{Y}$ is the semantic label of $\mathbf{x}$.

In practice, we take the well-trained BigBiGAN [39] generator as the data source, which was trained jointly with an additional encoder to learn the inverse mapping as BiGAN [41] and ALI [42]. It has been proved that these models have the following property.[2]

**Lemma 3.3** (Theorem 2 in [41]). *Assume discriminator $D$, generator $G$ and encoder $E$ are optimal, if $G$ and $E$ are deterministic, then $G^{-1} = E$ almost everywhere.*

Lemma 3.3 ensures the invertibility of the pretrained generator $G$. In COP-Gen (Figure 1c), the invertibility ensures $G(T_\mathbf{z}(\mathbf{z}))$ can be mapped back to a deterministic place $T_\mathbf{z}(\mathbf{z})$ in the latent space. Thus training a navigator $T_\mathbf{z} : \mathbf{z} \to T_\mathbf{z}(\mathbf{z})$ can make sense. Otherwise the navigator would be confused if there were more than one destination. The formal proof will be given in Theorem 3.4. We also use a pretrained flow model [43], which is naturally invertible, as the data source and conduct a toy experiment in Appendix D.

---

[2]It may not hold for other typical GAN generators.

**Theorem 3.4** (Optimal $T_{\mathbf{z}}$). *Under the above assumption and lemma, given specific $G(\mathbf{z})$, $T_{\mathbf{z}}^*$ obtained by optimizing Eq.* (5) *generates optimal positives $G(T_{\mathbf{z}}^*(\mathbf{z}))$ for contrastive learning.*

*Proof.* Below we use the notations in Figure 1c, i.e., $\mathbf{z}' = T_{\mathbf{z}}(\mathbf{z}), \mathbf{x} = G(\mathbf{z}), \mathbf{x}' = G(\mathbf{z}'), \mathbf{v} = t_{\mathbf{x}}(\mathbf{x}), \mathbf{v}' = t_{\mathbf{x}}'(\mathbf{x}')$, for brevity. Using the chain rule of mutual information, the training objective in Eq. (5) can be written as

$$I(\mathbf{v}; \mathbf{v}') = I(\mathbf{v}; \mathbf{y}) - I(\mathbf{v}; \mathbf{y}|\mathbf{v}') + I(\mathbf{v}; \mathbf{v}'|\mathbf{y}). \tag{8}$$

We first prove that $\mathbf{v}$ introduces no extra semantic information when $\mathbf{v}'$ is known, i.e., the above second item

$$I(\mathbf{y}; \mathbf{v}|\mathbf{v}') = 0. \tag{9}$$

Strictly, the equality approximately holds within the error range of $\varepsilon$ as shown below.

Using the chain rule of mutual information, we have

$$I(\mathbf{y}; \mathbf{x}|\mathbf{v}') = I(\mathbf{y}; \mathbf{x}) + I(\mathbf{y}; \mathbf{v}'|\mathbf{x}) - I(\mathbf{y}; \mathbf{v}'). \tag{10}$$

Since $t_{\mathbf{x}}'$ preserves the semantics of $\mathbf{x}'$, and the semantic consistency between $\mathbf{z}$ and $\mathbf{z}' = T_{\mathbf{z}}(\mathbf{z})$ holds under the condition of $d(\mathbf{z}, T_{\mathbf{z}}(\mathbf{z})) \le \delta$ from Proposition 3.1, we have

$$|I(\mathbf{y}; \mathbf{x}) - I(\mathbf{y}; \mathbf{v}')| = |I(\mathbf{y}; \mathbf{x}) - I(\mathbf{y}; t_{\mathbf{x}}'(\mathbf{x}'))| = |I(\mathbf{y}; \mathbf{x}) - I(\mathbf{y}; \mathbf{x}')| \le \varepsilon \tag{11}$$

by combining Eq. (7) and (4). In addition, under Lemma 3.3, $\mathbf{x}'$ and thus $\mathbf{v}' = t_{\mathbf{x}}'(\mathbf{x}')$ can be expressed as a function of $\mathbf{x}$, so that

$$I(\mathbf{y}; \mathbf{v}'|\mathbf{x}) = 0. \tag{12}$$

By plugging Eq. (11) and (12) into Eq. (10), we find $I(\mathbf{y}; \mathbf{x}|\mathbf{v}')$ is approximately zero within the error range of $\varepsilon$. And $\mathbf{v} = t_{\mathbf{x}}(\mathbf{x})$ is a function of $\mathbf{x}$. Therefore, $\mathbf{y} \to \mathbf{v}' \to \mathbf{x} \to \mathbf{v}$ forms a Markov chain approximately. Therefore, Eq. (9) holds within the error range of $\varepsilon$.

Now, Eq. (8) becomes

$$I(\mathbf{v}; \mathbf{v}') = I(\mathbf{v}; \mathbf{y}) + I(\mathbf{v}; \mathbf{v}'|\mathbf{y}). \tag{13}$$

Since $t_{\mathbf{x}}$ is semantic-preserving, the first item $I(\mathbf{v}; \mathbf{y})$ in Eq. (13) equals to the constant $I(\mathbf{x}; \mathbf{y})$. Therefore, minimizing Eq. (13) will make the second item, which represents the nuisance information, tend to zero. Formally, under the semantic constraint, $T_{\mathbf{z}}^*$ obtained by minimizing Eq. (13) satisfies

$$I(\mathbf{v}; \mathbf{v}'^*) \to I(\mathbf{x}; \mathbf{y}), \tag{14}$$
$$I(\mathbf{v}; \mathbf{v}'^*|\mathbf{y}) \to 0, \tag{15}$$
$$\text{s. t. } d(\mathbf{z}, T_{\mathbf{z}}^*(\mathbf{z})) \le \delta,$$

within the semantic error $\varepsilon$ defined in Proposition 3.1, where $\mathbf{v}'^* = t_{\mathbf{x}}'(G(T_{\mathbf{z}}^*(\mathbf{z})))$. According to Definition 2.1, $T_{\mathbf{z}}^*$ creates optimal positives for contrastive learning in latent space. □

## 3.3 Practical Details

**Choice of $T_{\mathbf{z}}$.** The latent space navigator $T_{\mathbf{z}}$ is implemented as a nonlinear two-layer neural network $T_{\mathbf{z}}(\mathbf{z}) = \boldsymbol{W}_2(\sigma(\boldsymbol{W}_1\mathbf{z} + \boldsymbol{b}_1)) + \boldsymbol{b}_2$ by default, where $\sigma$ is the ReLU [44] function. We adopt nonlinear trajectories in latent space since linear trajectories may cause generated images distorted or meaningless [45, 46]. Linear $T_{\mathbf{z}}$ is also experimented as an ablation in Section 4.3.

**Implementation of $d(\mathbf{z}, \mathbf{z}') \le \delta$.** In practice, we model $\mathbf{z}'$ as $\mathbf{z}' = \mathbf{z} + T_{\mathbf{z}}(\mathbf{z})$ instead of $\mathbf{z}' = T_{\mathbf{z}}(\mathbf{z})$, since random initializing the parameters of $T_{\mathbf{z}}$ may not guarantee $d(\mathbf{z}, \mathbf{z}') \le \delta$ thus violate the semantic constraint. Take L2 distance as an example, $d(\mathbf{z}, \mathbf{z}')$ now becomes $d_2(\mathbf{z}, \mathbf{z}') = \sqrt{\sum_{k=1}^{K} (T_{\mathbf{z}}(\mathbf{z})_k)^2}$. Since $\mathbf{z} \in \mathbb{R}^K$ is usually sampled from an isotropic Gaussian $\mathcal{N}(\mathbf{0}, \mathbf{I})$, $T_{\mathbf{z}}(\mathbf{z})$ approximately (due to nonlinear $T_{\mathbf{z}}$) follows Gaussian distribution with marginals $T_{\mathbf{z}}(\mathbf{z})_k \sim \mathcal{N}(\mu_k, \sigma_k^2)$, where $\mu_k$ and $\sigma_k$ are determined by $\boldsymbol{W}_1, \boldsymbol{W}_2, \boldsymbol{b}_1, \boldsymbol{b}_2$. Thus $\mathbb{E}[d_2(\mathbf{z}, \mathbf{z}')^2] = \sum_{k=1}^{K} \mathbb{E}[(T_{\mathbf{z}}(\mathbf{z})_k)^2] = \sum_{k=1}^{K} (\mu_k^2 + \sigma_k^2)$. Therefore, in order to satisfy $d(\mathbf{z}, \mathbf{z}') \le \delta$ experimentally, we control the initialization of the parameters in $T_{\mathbf{z}}$ and monitor the mean and standard deviation of $T_{\mathbf{z}}(\mathbf{z})$ as well as the quality of generated images during training. A summary of the implementation of the semantic consistency constraint during the whole training process is discussed in Appendix B.

# 4 Experiments

In this section we conduct experiments on the proposed COP-Gen approach and compare it with other latent transformation and real-image-based methods. We use Pytorch [47] for all experiments.

## 4.1 Experimental Settings

**Pretrained Generative Models.** Most of our self-supervised experiments are conducted using ImageNet ILSVRC-2012 [38] pretrained BigBiGAN [39, 48]. We do not use the encoder of BigBiGAN, but only use the generator as a black-box data source. Besides, some additional self-supervised experiments leveraging MNIST [49] pretrained Residual Flow [43] model can be found in Appendix D. We also extend our approach to supervised contrastive learning [50] in Appendix E.

**COP-Gen Training.** The latent space navigator $T_{\mathbf{z}}$ is implemented as a nonlinear two-layer MLP by default. The sizes of the input, hidden and output layers are the same, all equal to the latent dimension (120-d) of BigBiGAN. Data space SimCLR [11] augmentation is adopted, which is implemented to be differentiable by applying `torchvision.transforms` to tensors $\mathbf{x}$ on-the-fly, to let gradients flow from $\mathbf{v} = T_{\mathbf{x}}(\mathbf{x})$ to $\mathbf{x}$. The backbone of the mutual information estimator $f$ is ResNet-18 [51] to save GPU memory. We use InfoNCE loss [9] as the adversarial training objective for $f$ and $T_{\mathbf{z}}$, with the temperature set to 0.1. Adam [52] ($\beta_1 = 0.5, \beta_2 = 0.999$) is used as the optimizer, where the learning rate is set to $3 \times 10^{-5}$ for $f$ and $1 \times 10^{-5}$ for $T_{\mathbf{z}}$. We train over $200K$ generated samples[3] with batch size of 176, which takes about 1 hour on 4 NVIDIA 2080 Ti GPUs.

**Methods for Comparison.** We compare our COP-Gen with the following positive generation methods. The baseline **SimCLR** [11], which leverages data space transformation (e.g., random cropping, color jittering, etc.) to create positive pairs. **InfoMin Aug.** [20], which is inspired by the InfoMin principle and adopts additional (cf. SimCLR Aug.) data augmentations. **NNCLR** [25], which uses nearest neighbors in the learned representation space as the positives. **Negative Feature Transformation** [53], which creates hard negatives through feature interpolation. Two latent space transformation methods introduced in GenRep [28]. The one is through **Gaussian** transformation, with standard deviation set to 0.2 determined by grid searching. The other is a "**steering**" method introduced in [54], which learns a shared latent trajectory over $160K$ samples to mimic data space transformations (including zoom, shifts, rotations, and color transformations), and then randomly samples the length of the learned trajectory to create positives. More implementation details are given in Appendix C.1.

**Comparison of Computational Overhead.** When comparing with non-learning-based latent transformation methods (GenRep Gaussian [28]), our COP-Gen introduces additional navigator training overhead. This cost also happens in the previous learning-based latent "steering" method [28]. When comparing with real-image-based baseline [11], all computations including navigator training and positive sampling in the latent space of generative models are considered. All these computations are not too much (several GPU hours). But if we take training generative models into consideration, the extra training cost would be expensive (typically several GPU weeks). Nevertheless, the community has released many well-trained generative models for researchers to use. The detailed computational overhead is listed in Appendix F.

**Contrastive Learning.** We conduct contrastive learning on both real and synthetic datasets. For both of them in all experiments, we use $128 \times 128$ image size, which is the output resolution of the pretrained BigBiGAN generator. The real image encoder is trained on unlabeled ImageNet [38] which has $\sim 1.28M$ images of 1000 classes (ImageNet-1K), and the synthetic image encoder is trained on $1.3M$ randomly sampled anchor images from BigBiGAN. We use InfoNCE as the loss, which is optimized using SGD with momentum of 0.9, the learning rate of $0.03 \times \text{BatchSize}/256$, and weight decay of $10^{-4}$. We train with batch size of 224 for 100 epochs and decay the learning rate using the cosine decay schedule [55], which takes $\sim 3$ days for ResNet-50 on 2 NVIDIA 2080 Ti GPUs.

---

[3]See more discussions on the number of training samples (i.e. when to terminate the COP-Gen training process) in Section 4.3 and Appendix B.

Table 1: ImageNet-1K linear evaluation and PASCAL VOC object detection with different self-supervised contrastive positive generation methods. All the reported are our reproduced results.

| Method | $T_{\mathbf{z}}$ | $T_{\mathbf{x}}$ | ImageNet Linear Evaluation | | VOC Object Detection | | |
| | | | Top-1 | Top-5 | AP | $AP_{50}$ | $AP_{75}$ |
|---|---|---|---|---|---|---|---|
| *Training on ImageNet-1K real images:* | | | | | | | |
| SimCLR [11] | N/A | SimCLR Aug. | 49.44 | 75.58 | 52.91 | 78.68 | **58.51** |
| Neg FT [53] | N/A | SimCLR Aug. | 50.82 | 75.09 | 51.08 | 77.73 | 55.05 |
| InfoMin [20] | N/A | InfoMin Aug. | 50.93 | 75.87 | 51.11 | 77.58 | 55.55 |
| NNCLR [25] | N/A | SimCLR Aug. | 51.97 | 76.80 | 51.70 | 77.93 | 56.27 |
| *Training on BigBiGAN synthetic images:* | | | | | | | |
| GenRep [28] | None | SimCLR Aug. | 41.63 | 66.57 | 51.05 | 77.22 | 55.77 |
| NNCLR [25] | None | SimCLR Aug. | 42.46 | 66.46 | 50.40 | 76.74 | 55.00 |
| InfoMin [20] | None | InfoMin Aug. | 42.83 | 67.52 | 50.18 | 75.68 | 54.71 |
| Neg FT [53] | None | SimCLR Aug. | 43.26 | 67.21 | 50.45 | 77.11 | 54.88 |
| GenRep [28] | Gaussian | SimCLR Aug. | 48.73 | 73.13 | 50.20 | 77.03 | 54.37 |
| GenRep [28] | Steering | SimCLR Aug. | 51.19 | 74.97 | 51.42 | 77.96 | 56.43 |
| COP-Gen (ours) | Optimal | SimCLR Aug. | **53.25** | **77.16** | **53.09** | **78.95** | 57.99 |

## 4.2 Experimental Results

**Linear Evaluation on ImageNet-1K.** We train a fully connected linear classifier on top of the frozen 2048-d embeddings from the pretrained ResNet-50 encoder. It is trained using SGD with momentum of 0.9, batch size of 224 for 60 epochs with cosine learning rate decay. Following [28], the initial learning rate is set as $0.3 \times \text{BatchSize}/256$ for the real and $2 \times \text{BatchSize}/256$ for the synthetic respectively. We report Top-1 and Top-5 classification accuracies on ImageNet-1K validation set.

The comparison result is presented in Table 1.[4] First, COP-Gen achieves the best performance compared to other synthetic-image-based methods, demonstrating the effectiveness of our latent space optimal positive generation approach. In addition, it is worth noting that synthetic-image-based contrastive representation learning can outperform the real-image-based method, as long as the proper latent transformation method is adopted. That is because the latent transformation brings more diversity to and could achieve instance-wise optimal positives. Although there is still a gap in the resolution and quality between synthetic and real images, this phenomenon inspires us to further develop black-box generative models as datasets to perform visual representation learning.

**Object Detection on PASCAL VOC.** We test the pretrained encoders on the downstream object detection task to evaluate the feature transferability. Following the protocol in [10], we use `detectron2` [56] to train a Faster-RCNN [57] with the R50-C4 backbone. We fine-tune all layers end-to-end with batch size of 4 for $96K$ iterations (~23 epochs) on the PASCAL VOC [58] `trainval07+12` set and evaluate on the `test07` set. The results are in Table 1. It can be seen that COP-Gen achieves consistent gains over other synthetic encoders and competitive performance compared with real-image-based baselines.

**Transfer Learning and Semi-Supervised Learning Evaluations.** Following the procedure in [11, 25], we evaluate the pretrained encoders on transfer learning and semi-supervised learning tasks. For transfer learning task, we train a linear classifier on following datasets: Food101 [59], CIFAR10 [60], CIFAR100 [60], SUN397 [61], Pets [62], Caltech-101 [63], and Flowers [64]. For semi-supervised learning task, we follow the setting in [11], i.e., sample 1% or 10% of the labeled ImageNet training set in a class-balanced way, and fine-tune the whole network on the labeled data. These two experiments are implemented based on the open-source codebase `VISSL` [65] using its standard set of hyper-parameters and evaluation protocols. We compare our method with the baseline SimCLR and two of the most competitive methods: NNCLR on the real and latent "steering" method. Table 2 shows the results. It is worth noting that synthetic-image-trained encoders outperform real-image-trained encoders on nearly all transfer and semi-supervised benchmarks. And our method achieves competitive performances when compared with the previous latent "steering" method.

---

[4] The gap between our reproduced results and those on real images is because we use smaller image size, batch size and different optimizers. All these details are clarified in Section 4.1. Results with standard deviations after running experiments multiple times and longer contrastive learning are given in Appendix C.3.

Table 2: Comparison of transfer learning and semi-supervised learning performance. For transfer classification we report Top-1 accuracy. For semi-supervised learning we report Top-5 accuracy.

| Method | Linear Transfer Classification | | | | | | | Semi-Supervised Label Fraction | |
| | Food101 | CIFAR10 | CIFAR100 | SUN397 | Pets | Caltech-101 | Flowers | 1% | 5% |
|---|---|---|---|---|---|---|---|---|---|
| *Training on ImageNet-1K real images:* | | | | | | | | | |
| SimCLR [11] | 65.16 | **84.44** | 60.99 | **66.26** | 69.72 | 81.96 | 82.81 | 45.77 | 77.34 |
| NNCLR [25] | 62.53 | 84.10 | 60.43 | 64.49 | 70.35 | 82.45 | 82.84 | 44.09 | 76.28 |
| *Training on BigBiGAN synthetic images:* | | | | | | | | | |
| GenRep Steering [28] | 66.17 | 84.23 | **63.29** | 62.46 | 69.94 | **84.65** | **84.78** | **52.33** | 78.70 |
| COP-Gen (ours) | **67.05** | 84.09 | 62.44 | 63.82 | **73.56** | 84.63 | 84.36 | 51.84 | **78.81** |

## 4.3 Ablation Study

In this section we present a thorough analysis of COP-Gen and its influence on contrastive learning.

**Default Setting.** We conduct ablation experiments on ImageNet-100, a subset of ImageNet-1K, as done in [66, 15, 28]. In this setting, the synthetic encoder is trained on $130K$ anchor images sampled from pretrained BigBiGAN. Note that these sampled images actually come from 1000 classes since the fixed BigBiGAN was trained on unlabeled ImageNet-1K. Therefore, we here only perform contrastive learning on these synthetic images and report ImageNet-100 linear evaluation results, since training a real encoder on ImageNet-100 will be an unfair comparison. Here, the number of contrastive pretraining epoch is set to 200, and other training hyperparameters and process are the same as those in Section 4.1 by default. The ImageNet-100 linear evaluation results of the default setting are presented in the upper part of Table 3.

Table 3: ImageNet-100 linear evaluation on pretrained encoders based on synthetic images of different ways of positive generation.

| Method | $T_\mathbf{z}$ | $T_\mathbf{x}$ | Top-1 |
|---|---|---|---|
| *Default setting:* | | | |
| GenRep [28] | None | SimCLR Aug. | 58.26 |
| GenRep [28] | Gaussian | SimCLR Aug. | 62.62 |
| GenRep [28] | Steering | SimCLR Aug. | 63.08 |
| COP-Gen (ours) | Optimal | SimCLR Aug. | **63.96** |
| *Contrastive pretraining w/o data augmentation:* | | | |
| GenRep [28] | Gaussian | None | 28.90 |
| GenRep [28] | Steering | None | 28.68 |
| COP-Gen (ours) | Optimal | None | **37.40** |

**Do We Still Need Data Augmentation $T_\mathbf{x}$?** Having obtained the optimal latent space transformation $T_\mathbf{z}$, a natural question is that do we still need data augmentation $T_\mathbf{x}$ *when performing contrastive learning*. We answer this question by removing $T_\mathbf{x}$ and using only $T_\mathbf{z}$ to create positive pairs. The results are shown in the bottom part of Table 3. It can be seen that in the absence of $T_\mathbf{x}$, the performance drops significantly, even though $T_\mathbf{z}$ obtained by our COP-Gen is optimal. And best result is achieved by combining instance-specific latent (COP-Gen) and data space transformations.

Further, we conduct ablations by removing color, crop, and all transformations in the data space respectively *at both navigator learning and contrastive learning stages*, to see whether the learned $T_\mathbf{z}(\cdot)$ (together with the frozen $G$) could cover these data space transformations. As shown in Table 4, the large performance drops imply that $G(T_\mathbf{z}(\cdot))$ has poor ability to cover color and crop augmentations, which are crucial when creating contrastive pairs [11]. The reason behind this is probably because images generated by GANs are usually object-centric and color-biased [40]. To overcome this, one might involve these complex $T_\mathbf{x}$ when training generative models $G$, or train $G$ together with our navigator $T_\mathbf{z}$, so that $G(T_\mathbf{z}(\cdot))$ could cover more $T_\mathbf{x}$.

Table 4: Ablation on $T_\mathbf{x}$. Navigator training and contrastive learning adopt the *same* policy.

| $T_\mathbf{x}$ | SimCLR Aug. | w/o color | w/o crop | None |
|---|---|---|---|---|
| Top-1 | **63.96** | 55.26 | 43.84 | 36.16 |

**Number of Training Samples for $T_\mathbf{z}$.** By default, we decide when to terminate COP-Gen training process by monitoring the quality of generated positives. This allows efficient experiments in Section 4.2 without operations like grid searching. Now, we increase the number of training samples to investigate its influence on the generated positives and contrastive learning performance. For random anchor images, we show how the generated positives evolve over the training time in the middle part of Figure 3.

It can be seen that: (1) In the first row, the clothes of the person change and the background becomes simpler and simpler. On the contrary, for an anchor with simple background (row 2), the training process makes it gradually more complex. This phenomenon shows that the learned positives are instance-specific. In addition, the COP-Gen training process only changes the above nuisance information between the positive pair, while keeping the semantics (the person and the obelisk) unchanged, so that the following contrastive learning stage will not focus on the semantic-irrelevant information. (2) As for the last positive in row 4, the semantics of the subject is changed (a human face appearing on the cup), which indicates that the semantic consistency constraint is violated and the training process should have been terminated.

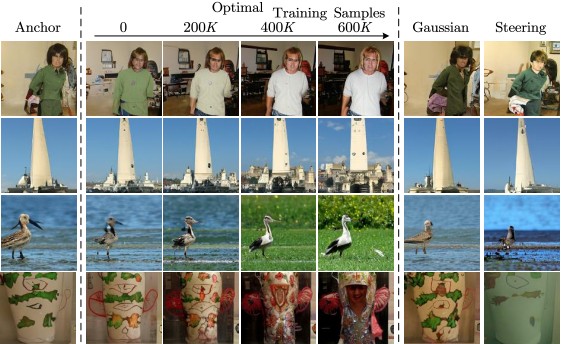

Figure 3: Middle: Generated anchor-specific positives by our COP-Gen approach with respect to the number of training samples. A smoother evolution with more examples is given in Figure 4 in Appendix. Right: Results by other latent-transformation-based positive generation methods.

The contrastive learning results using these positives generated at different training periods are shown in Table 5. Note that although the best performance may not be achieved through simply and conservatively monitoring the quality of generated positives, the results only fluctuate slightly and all of them outperform other non-instance-specific latent transformation methods.

Table 5: Ablation on different numbers of COP-Gen training samples.

| Training Samples | $200K$ | $400K$ | $600K$ |
|---|---|---|---|
| Top-1 | 63.96 | **64.12** | 63.56 |

**Linear** $T_{\mathbf{z}}$**.** We also conduct experiments on modeling $T_{\mathbf{z}}$ as a linear trajectory. With such design, we achieve $61.18\%$ Top-1 accuracy, which falls behind non-linear $T_{\mathbf{z}}$ and other latent transformation methods. This may be due to the limited modeling capability of linear transformations.

### 4.4 Visualization Analysis

We give a visual comparison on the generated positives with other latent transformation methods. As shown in Figure 3, the "steering" method [54, 28], which learns a shared latent trajectory with random scales to mimic data augmentations, causes unrealistic in a few images. This is partly because the pretrained GANs could not achieve these strong data space transformations. Relatively, the Gaussian transformation constructs positives in a conservative way after grid searching its standard deviation [28], which only ensures the semantic consistency but lacks diversity. In contrast, our COP-Gen is aware of the instance-specific nuisance information and removes that between the positives.

## 5 Related Work

**Contrastive Learning.** The goal of contrastive learning is to contrast positive with negative pairs [6]. Without access to labels, positive pairs are typically created from a single image through data augmentation while negative pairs are sampled randomly from the whole dataset [10, 11]. Many works have been proposed to improve this strategy by correcting the sampling bias or mining hard samples. For negative pairs, some works [14, 19] eliminate the false negatives in randomly sampled pairs. Some works construct hard negatives from existing samples [15, 24, 17, 16, 53], or adversarial perturbations [21, 18]. For positive pairs, Chen et al. [11] and Tian et al. [20] revisit common data augmentations and Reed et al. [67] searches augmentation policies. To construct hard positives, feature extrapolation [53] and neighbor discovery [24, 25, 26, 27, 68] are adopted.

Our goal is more than mining hard but to create optimal positives. The most relevant works to ours are [69, 20] that share the same high level idea to construct positives with minimum sufficient information. However, there are several fundamental differences. The work [69] focuses on graph contrastive learning and the model design relies on domain knowledge, while our target is general visual representation learning without any supervision. Besides proposing the InfoMin principle,

Tian et. al [20] also conducted a view learning experiment to find optimal positives in data space. However, the construction is under the color-channel splitting framework and requires the labels of downstream task. In contrast, COP-Gen finds instance-specific optimal positives in the latent space of pretrained unconditional generative models, which is a fully unsupervised setting.

Besides, the objective in [70, 71] simply pulls positives together without explicitly contrasting with negative samples. We check if it is possible to combine our contrastive optimal positive generation method with these non-contrastive frameworks in Appendix C.4.

**Using Generative Models as Data Source.** Deep generative models have been developed to produce photo realistic images [36, 37, 72, 73], thus they can be used as great tools for generating desired image datasets. Some works investigate intermediate GAN representations to construct part segmentation datasets [74, 75], while some treat pretrained GANs as black-box models and use them to augment data for robustness [76, 35] or ensembling [77]. The pretrained generative models can be data source alone. Some works [78, 79] leverage class-conditional generative models to train a classifier, while GenRep [28] and this paper focus on a more general setting of visual representation learning.

**Latent Space Navigation.** A key technique for using pretrained generative models to produce interested images is latent space navigation [32, 33, 80, 81, 46, 82, 83], which manipulates images by discovering interpretable directions in the latent space. Different from these works, our goal is to learn not only the direction but also the magnitude of the trajectory. In addition, the trajectory is instance-specific, i.e., the learned transformations applied to different images have different meanings. Refer to Figure 3 for the visualization result.

# 6 Discussion

In this paper, we propose an instance-specific contrastive positive generation approach based on well-trained generative models. Thanks to the attractive property of the latent space, we simply optimize a latent trajectory that reduces as much information as possible within a reasonable magnitude. Theoretical analysis and experimental results demonstrate the effectiveness of our approach.

**Limitations and Broader Impact.** Since this work uses synthetic data to perform contrastive learning, the performance naturally and partly depends on the quality of pretrained generative models. Nevertheless, having seen that learning from synthetic dataset has outperformed learning from the real, which might impact representation learning community as a whole, we are inspired to further develop more proper generative models (e.g., to reduce the reliance on data space augmentations) as dataset sources to perform contrastive learning.

Besides, learning from synthetic data alleviates several societal issues of learning from real datasets, which might be private [84] or biased [85]. On the other hand, pretrained generative models may still leak [86] or inherit the biases from the training data [87]. These risks and impacts on representation learning need to be considered when leveraging pretrained generative models as the data source.

Take a class-imbalanced setting for example. Although vanilla contrastive learning can learn more balanced feature space than its supervised counterpart since it does not use labels [88], it still could not immune to imbalance. It is hypothesized that the high-frequency "classes" might dominate the learning process and leaves the low-frequency "classes" under-learned [89]. We here hypothesize that the learning of samples from low-frequency "classes" might benefit from COP-Gen generated positives. The training process of COP-Gen has seen the overall imbalanced distribution thus has the potential to take it into consideration to create more proper positives.

# Acknowledgements

This work is partially supported by National Key R&D Program of China (No. 2017YFA0700800), and National Natural Science Foundation of China (NSFC): 61976203, 61876171. We thank Dr. Xiaoyi Yin for valuable suggestions on the theoretical part of this work. We also thank anonymous reviewers for their constructive feedback.

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
