# A  The Training Algorithm of COP-Gen

Algorithm 1 summarizes the proposed COP-Gen approach.

---
**Algorithm 1** COP-Gen training algorithm.

---
**Input:** pretrained generative model $G(\cdot)$, prior distribution $p_{\mathbf{z}}$, batch size $K$, initialized latent space navigator $T_{\mathbf{z}}(\cdot)$, data space augmentations $T_{\mathbf{x}}$, encoder $f(\cdot)$
**Output:** trained latent space navigator $T_{\mathbf{z}}(\cdot)$
**for** number of training iterations **do**
    **for** $k = 1 \, to \, K$ **do**
        sample $\mathbf{z}_k$ from noise prior $p_{\mathbf{z}}$
        $\mathbf{z}'_k = T_{\mathbf{z}}(\mathbf{z}_k)$                      # latent transformation
        **if** $d(\mathbf{z}_k, \mathbf{z}'_k) > \delta$              # semantic constraint is violated
            **return** $T_{\mathbf{z}}(\cdot)$, and discard $f(\cdot)$
        $\mathbf{x}_k = G(\mathbf{z}_k), \mathbf{x}'_k = G(\mathbf{z}'_k)$        # generate images
        draw two data augmentations $t_{\mathbf{x}} \sim T_{\mathbf{x}}, t'_{\mathbf{x}} \sim T_{\mathbf{x}}$
        $\mathbf{v}_k = t_{\mathbf{x}}(\mathbf{x}_k), \mathbf{v}'_k = t'_{\mathbf{x}}(\mathbf{x}'_k)$     # data transformation
        $\mathbf{h}_k = f(\mathbf{v}_k), \mathbf{h}'_k = f(\mathbf{v}'_k)$        # representations
    **end for**
    # calculate InfoNCE loss for minibatch
    **define** $\mathcal{L}_k = -\log \frac{\exp(\text{sim}(\mathbf{h}_k, \mathbf{h}'_k))}{\sum_{j=1}^{K} \mathbb{1}_{[j \neq k]} \exp(\text{sim}(\mathbf{h}_k, \mathbf{h}_j)) + \sum_{j=1}^{K} \exp(\text{sim}(\mathbf{h}_k, \mathbf{h}'_j))}$
    $\mathcal{L} = \frac{1}{K} \sum_{k=1}^{K} \mathcal{L}_k$
    update encoder $f$ to minimize $\mathcal{L}$
    update latent space navigator $T_{\mathbf{z}}$ to maximize $\mathcal{L}$
**end for**
**return** $T_{\mathbf{z}}(\cdot)$, and discard $f(\cdot)$

---

# B  Implementation of Semantic Consistency Constraint

The semantic consistency constraint, i.e. $d(\mathbf{z}, \mathbf{z}') \leq \delta$, needs to be held during the whole training process of COP-Gen.

**Initialization of $\mathbf{z}'$.** In practice, $\mathbf{z}'$ is modeled as $\mathbf{z}' = \mathbf{z} + T_{\mathbf{z}}(\mathbf{z})$ instead of $\mathbf{z}' = T_{\mathbf{z}}(\mathbf{z})$, since random initializing the parameters of $T_{\mathbf{z}}$ may not guarantee $d(\mathbf{z}, \mathbf{z}') \leq \delta$ thus violate the semantic constraint. In addition, for the initialization of parameters in the $T_{\mathbf{z}}$ (MLP), we use the method introduced by [90], which satisfies the semantic consistency at the beginning of the training process and works well.

**Termination of the training process.** By default, we decide the termination of COP-Gen training process by watching the image quality of generated positives, as shown in Figure 3 and discussed in Section 4.3. In addition, as introduced in Section 3.3, we also monitor the mean and standard deviation of all trajectories in the batch and compare them with the prior value investigated in [28]. This is used as an auxiliary monitor signal. A more elegant way is to terminate automatically. To this end, one might introduce a metric for calculating the semantic consistency between $G(T_{\mathbf{z}}(\mathbf{z}))$ and $G(\mathbf{z})$. The time when it drops dramatically indicates the termination. LPIPS metric [91] or the feature similarity of pretrained powerful CLIP encoder [92] applied on the triplet $\{G(\mathbf{z}), G(T_{\mathbf{z}}(\mathbf{z})_{init}), G(T_{\mathbf{z}}(\mathbf{z})_{curr})\}$ might be reasonable choices and we leave it as the future work.

# C  More Details and Results of Self-Supervised ImageNet Experiments

## C.1  Implementation Details of Reproduced Baselines

We use the code[5] (MIT license) of GenRep [28] to implement contrastive learning (both our method and **GenRep baselines**). We follow their experimental setting and hyper-parameters as explained in the last paragraph of Section 4.1 in the main text, except that we perform 100 and 200 (see Appendix C.3) epochs of contrastive pretrainig on ImageNet-1K to get more convincing results, in contrast to 20 epochs in GenRep.

---
[5] https://github.com/ali-design/GenRep

The reproduction of InfoMin Aug. [20], NNCLR [25], and Neg FT [53] adopts the same setting for fair comparisons where their hyper-parameters (learning rates in practice) are grid searched to make them work. **InfoMin Aug.** is implemented based on the released code[6], where additional data augmentations (cf. SimCLR Aug.) are adopted. **NNCLR** uses nearest neighbors in the learned representation space as the positives, and is implemented based on the open source codebase `Lightly`[7] [93] (MIT license). **Neg FT** creates hard (diverse) negatives through feature interpolation, and is implemented based on the released code[8].

## C.2 More Visualization Results

Figure 4 in this subsection gives a smoother version of Figure 3 in the main text, where more examples and training steps are shown for visualizing the evolution of generated anchor-specific positives by COP-Gen for self-supervised contrastive learning.

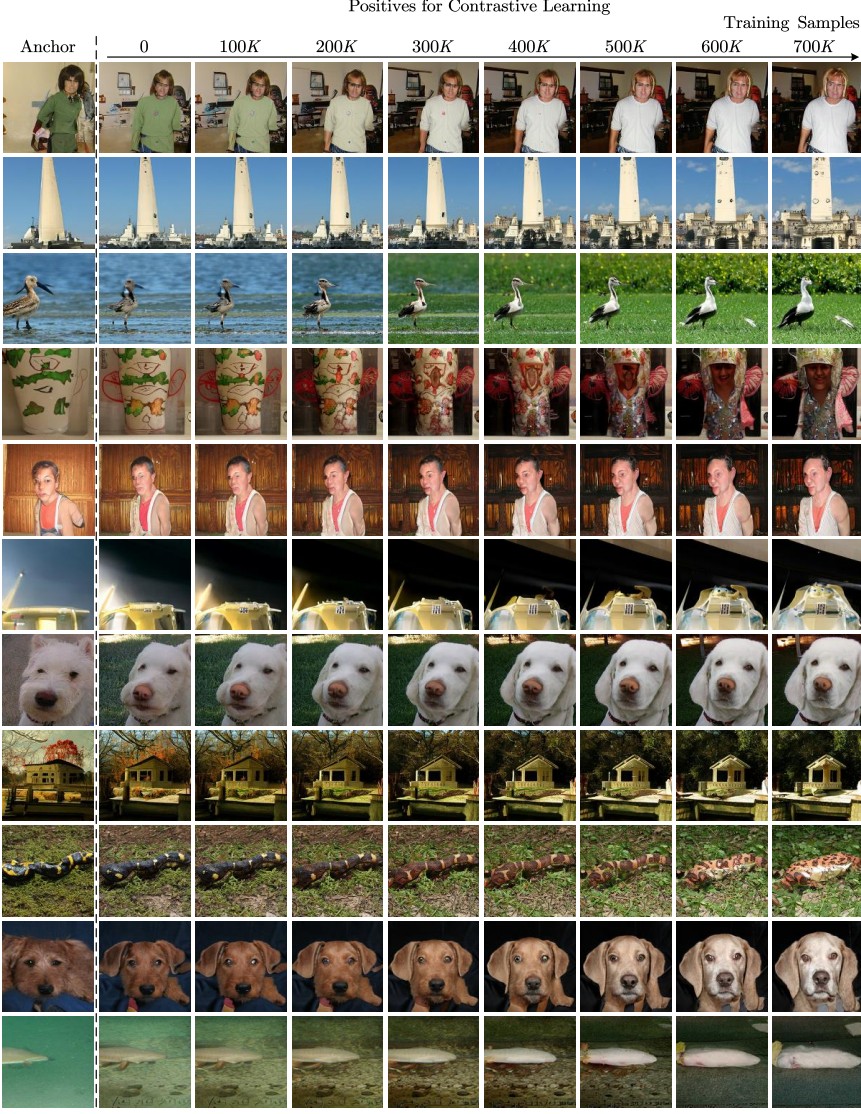

Figure 4: Generated anchor-specific positives using ImageNet-1K pretrained unconditional BigBi-GAN by our COP-Gen approach **for self-supervised contrastive learning** with respect to the training steps.

---

[6] https://github.com/HobbitLong/PyContrast
[7] https://docs.lightly.ai/examples/nnclr.html
[8] https://github.com/DTennant/CL-Visualizing-Feature-Transformation

## C.3 Longer Contrastive Learning and Experiments with Multiple Runs

In this subsection, we perform contrastive learning over 200 epochs under the ImageNet-1K setting, to see whether our method could benefit from longer training epochs. We compare our COP-Gen with the baseline SimCLR [11] and the most competitive NNCLR [25]. All these experiments are repeated for three times. For our method, the repetition is also conducted at the navigator training (COP-Gen) stage.

The results with standard deviations are reported in Table 6. Although COP-Gen has relatively higher standard deviations since the termination of navigator training is decided by simply monitoring the quality of generated positives, it achieves overall good performance against the state-of-the-art real-image-based method NNCLR. As for longer pretraining, NNCLR could benefit more from that, since the "true positive rate" of nearest neighbours in the learned representation space increases gradually as the training goes. Nevertheless, our method, fed with the learned optimal positive pairs, converges quickly in contrastive pretraining (achieves competitive performance at 100 epochs).

Table 6: Top-1 accuracy of ImageNet-1K linear evaluation with respect to different contrastive learning epochs. All the reported are our reproduced results.

| Method | Pretraining Data Source | $T_\mathbf{z}$ | $T_\mathbf{x}$ | Top-1 w.r.t. Pretraining Epochs | |
| --- | --- | --- | --- | --- | --- |
| | | | | 100 | 200 |
| SimCLR [11] | ImgaeNet-1K | N/A | SimCLR Aug. | 49.37±0.07 | 51.89±0.39 |
| NNCLR [25] | ImgaeNet-1K | N/A | SimCLR Aug. | 52.18±0.19 | **53.88**±0.25 |
| COP-Gen (ours) | BigBiGAN | Optimal | SimCLR Aug. | **52.39**±0.85 | 52.73±0.65 |

## C.4 Combination with Non-Contrastive Frameworks

In this subsection, we check if our *contrastive* optimal positive generation method could be combined with *non-contrastive* frameworks. To do so we change the contrastive self-supervised learning (SSL) framework SimCLR to the non-contrastive BYOL [70], based on the open source codebase `Lightly`[9] [93]. We conduct experiments on ImageNet-100-scale BigBiGAN synthetic images and evaluate linear performance on ImageNet-100 (the setting in Section 4.3). Table 7 shows the results. It can be seen that changing the SSL framework has little impact on pretraining with synthetic dataset. When combining with non-contrastive BYOL, our COP-Gen could still improve the learned representations, which empirically indicates that *the optimal positive pairs for non-contrastive approaches might share the same InfoMin principle [20] with contrastive approaches*.

Table 7: Top-1 accuracy of ImageNet-100 linear evaluation with respect to different SSL frameworks.

| Method | SSL Framework | $T_\mathbf{z}$ | $T_\mathbf{x}$ | Top-1 |
| --- | --- | --- | --- | --- |
| GenRep [28] | SimCLR [11] | None | SimCLR Aug. | 58.26 |
| COP-Gen (ours) | SimCLR [11] | Optimal | SimCLR Aug. | **63.96** |
| GenRep [28] | BYOL [70] | None | SimCLR Aug. | 59.30 |
| COP-Gen (ours) | BYOL [70] | Optimal | SimCLR Aug. | **64.60** |

## C.5 Does the Quality of the Generative Model Affect the Final Result?

In theory, the learned representation can be improved with a perfect pretrained generative model as the data source. To simply check this, we use instance-conditioned GAN (IC-GAN) [94] as the data source for self-supervised contrastive learning. IC-GAN learns to model the distribution of the neighborhood of each data point, and achieves the state-of-the-art performance in Inception Score (IS) [95] and Fréchet Inception Distance (FID) [96] on unlabeled ImageNet generation. We conduct experiments on ImageNet-100-scale IC-GAN synthetic images and evaluate linear performance on ImageNet-100 (the setting in Section 4.3). For our COP-Gen, we train the navigator in its stored-instance-conditional sub-spaces, to ensure the semantic consistency between generated positives.

---

[9]https://docs.lightly.ai/examples/byol.html

Results compared with training on BigBiGAN generated images are presented in Table 8. Interestingly, for both GenRep baseline and our COP-Gen method, the learned representation does not benefit much from the "better" pretrained generative model. This might be because of the mismatched evaluation protocols between generative models' training and their usage in representation learning. We hope this could encourage the community to rethink the existing metrics of generative models. And how to design a prefect generative model for representation learning remains an open issue.

Table 8: Top-1 accuracy of ImageNet-100 linear evaluation with respect to different pretraining data sources. IS and FID scores are from [39, 94] respectively.

| Method | Pretraining Data Source | | | | $T_{\mathbf{z}}$ | $T_{\mathbf{x}}$ | Top-1 |
| | Generative Model | Resolution | IS ($\uparrow$) | FID ($\downarrow$) | | | |
|---|---|---|---|---|---|---|---|
| GenRep [28] | BigBiGAN [39] | 128 | 23.3±0.2 | 28.5±0.4 | None | SimCLR Aug. | 58.26 |
| GenRep [28] | IC-GAN [94] | 128 | **48.7**±0.2 | **11.7**±0.0 | None | SimCLR Aug. | 58.26 |
| COP-Gen (ours) | BigBiGAN [39] | 128 | 23.3±0.2 | 28.5±0.4 | Optimal | SimCLR Aug. | 63.96 |
| COP-Gen (ours) | IC-GAN [94] | 128 | **48.7**±0.2 | **11.7**±0.0 | Optimal | SimCLR Aug. | **64.06** |

# D   Residual Flow Experiments

In this section, we use MNIST [49] pretrained Residual Flow [43] as the dataset to perform self-supervised contrastive learning. Since the flow models strictly satisfy the invertibility like Lemma 3.3, they can be adopted as the pretrained generative models in our COP-Gen method.

**Contrastive Learning.** We perform contrastive learning using both real and synthetic images with image size of $28 \times 28$. The synthetic dataset contains the same number (60000) of anchor images with the real MNIST. Data space transformation $T_{\mathbf{x}}$ (including random $\pm 10°$ rotation, $\pm 10\%$ translation, and $0.9 \sim 1.1\times$ scaling) is used for both real and synthetic pretraining. The temperature of InfoNCE loss is set to $0.5$. We adopt ResNet-50 as the backbone and report Top-1 linear evaluation accuracy on MNIST test set. We conduct pretraining and linear evaluation over $100$ and $60$ epochs respectively, with batch size of $64$, initial learning rates of $0.1$ and $0.5$ for the real, $0.1$ and $1.5$ for the synthetic, taking about 1.5 hours in total on 2 NVIDIA 2080 Ti GPUs.

**Gaussian Baseline.** Following [28], we perform grid searching for the optimal standard deviation of the Gaussian transformation. We search over $std = [0.01, 0.05, 0.1, 0.2, 0.4]$ and find the optimal is $0.05$, which achieves the highest linear classification accuracy.

**COP-Gen Training.** The parameters of latent space navigator $T_{\mathbf{z}}$ are initialized to make the standard deviation of the output vector around $0.01$, in order to satisfy the semantic consistency (Proposition 3.1) at the beginning of the training process. The sizes of the input, hidden and output layers are all equal to the latent dimension (784-d) of the pretrained Residual Flow. The learning rate is set to $1 \times 10^{-6}$ for $f$ and $3 \times 10^{-7}$ for $T_{\mathbf{z}}$. We train over $400K$ generated samples with batch size of $64$, which takes about 8 hours on 4 NVIDIA 40GB A100 GPUs.

**Results.** The comparison result is shown in Table 9. Using the positives generated by COP-Gen outperforms the Gaussian transformation method. Unfortunately, COP-Gen falls slightly behind the real-image-trained encoder, probably due to the quality gap between the real and synthetic images.

Table 9: MNIST linear evaluation with respect to different ways of contrastive positive pair generation.

| Method | $T_{\mathbf{z}}$ | $T_{\mathbf{x}}$ | Top-1 |
|---|---|---|---|
| *Training on MNIST real images:* | | | |
| SimCLR [11] | N/A | ✓ | **99.3**5 |
| *Training on Residual Flow synthetic images:* | | | |
| GenRep [28] | None | ✓ | 98.54 |
| GenRep [28] | Gaussian | ✓ | 98.83 |
| COP-Gen (ours) | Optimal | ✓ | 98.95 |

# E Supervised Contrastive Learning

We extend our COP-Gen approach to supervised contrastive learning in this section. Following [28], we use the pretrained class-conditional BigGAN [36] as the data source. In Appendix E.1, we introduce an additional latent space based positive[10] generation baseline [28], and the theoretical differences of this approach compared with that of the self-supervised version in the main text. Next, we present the experimental results in Appendix E.2.

## E.1 Method

**Random Baseline.** To construct supervised contrastive positives using class-conditional generative models, Jahanian et al. [28] introduces an additional method that randomly draws a sample in the class-conditional sub-space as the positive. Intuitively, this strategy is quite sensible and optimal to some extent (Definition 2.1), since the positives are drawn independently and share the same class label. However, we will show in Appendix E.2.2 that this strategy could not identify and remove nuisance information in each anchor when compared with our COP-Gen.

**COP-Gen.** As for our supervised COP-Gen, there are two theoretical differences with the self-supervised version in the main text. **(1) Difference in Proposition 3.1.** The semantic consistency always holds in the supervised setting, since we perform COP-Gen latent transformation learning in the class-conditional sub-spaces of $G$, wherein two samples have the same label. **(2) No theoretical guarantee of Lemma 3.3.** Theoretically, the invertibility may not hold for the pretrained BigGAN here. As a consequence, the navigator would be confused since there might be more than one destination for a fixed starting point. Fortunately, we empirically find COP-Gen still learns meaningful transformations and outperforms other latent space transformation methods (see Appendix E.2.2).

## E.2 Experiments

### E.2.1 Experimental Settings

**Contrastive Learning.** We perform supervised contrastive learning using both real and synthetic images with image size of $128 \times 128$. For the synthetic data source, we use ImageNet-1K pretrained class-conditional BigGAN-deep-128 [36] with truncation set to 2.0, and randomly sample 1300 anchor images in 1000 classes (in total $1.3M$ to match the scale of ImageNet-1K). We use supervised contrastive loss (the default "out" version) [50] with the temperature set to $0.1$, which is optimized using SGD with momentum of 0.9, learning rate of $0.03 \times \text{BatchSize}/256$, and weight decay of $10^{-4}$. We train with batch size of 224 for 100 epochs and decay the learning rate using cosine decay schedule [55], which takes ~3 days for ResNet-50 on 2 NVIDIA 2080 Ti GPUs.

**COP-Gen Training.** We use supervised contrastive loss as the adversarial training objective for $f$ and $T_{\mathbf{z}}$, whose learning rates are both set to $1 \times 10^{-5}$. Note that one of the key successes of the supervised contrastive loss is due to introducing multiple positives with the same label in the batch (Table 1 of Sec. 7 in Appendix of [50]). Unfortunately, due to the limitation of GPU resources, we could not implement such large batch size of 6144 (or at least 2000) for 1000-class ImageNet, to make sure that each image has at least another one positive comes from a different instance. To alleviate this, we control the number of classes in the batch to be the half of the batch size (i.e., batch size is set to 500 with images sampled from 250 classes, to mimic 2000 batch size for 1000-class ImageNet). We train over $2M$ generated samples, which takes about 12 hours on 4 NVIDIA 40GB A100 GPUs.

**Methods for Comparison.** We compare our COP-Gen with the following positive generation methods. The baseline **SupCon** [50] method which leverages data space SimCLR augmentation to create positive pairs performed on both real and synthetic images. Three latent space transformation methods introduced in GenRep [28]: **Gaussian** transformation with standard deviation set to 1.0 here as in [28], the "**steering**" method same as the self-supervised version in Section 4.1, and the **random** method introduced in Appendix E.1.

---

[10] In this section, unless otherwise stated, "positive" refers to the other view of the anchor image, which is consistent with the self-supervised setting in the main text. We clarify this because in supervised contrastive learning [50], "positives" usually refer to all samples with the same label in the batch.

Table 10: ImageNet-1K linear evaluation and PASCAL VOC object detection with different supervised contrastive positive generation methods. All the reported are our reproduced results.

| Method | $T_{\mathbf{z}}$ | $T_{\mathbf{x}}$ | ImageNet Linear Evaluation | | VOC Object Detection | | |
| --- | --- | --- | --- | --- | --- | --- | --- |
| | | | Top-1 | Top-5 | AP | $AP_{50}$ | $AP_{75}$ |
| *Training on ImageNet-1K real images:* | | | | | | | |
| SupCon [50] | N/A | SimCLR Aug. | **66.40** | **87.57** | **52.18** | **79.15** | 56.77 |
| *Training on BigGAN synthetic images:* | | | | | | | |
| GenRep [28] | None | SimCLR Aug. | 55.93 | 78.78 | 51.49 | 79.09 | 55.65 |
| GenRep [28] | Gaussian | SimCLR Aug. | 53.67 | 77.00 | 45.57 | 74.56 | 48.07 |
| GenRep [28] | Steering | SimCLR Aug. | 55.82 | 78.47 | 49.35 | 76.86 | 53.76 |
| GenRep [28] | Random | SimCLR Aug. | 56.23 | 79.17 | 49.07 | 77.00 | 52.58 |
| COP-Gen (ours) | Optimal | SimCLR Aug. | 57.16 | 79.58 | 52.10 | 78.46 | **56.83** |

### E.2.2 Experimental Results

**Linear Evaluation and Object Detection.** In Table 10 we present the results for linear evaluation on ImageNet-1K and object detection on PASCAL VOC. The protocols and hyper-parameters are the same as in Section 4.2 in the main text. Among synthetic-image-trained encoders, COP-Gen ranks the first on linear evaluation and is on par with others on object detection, indicating that COP-Gen has produced more proper positive pairs for contrastive learning.

**Discussion on Class-Conditional Generative Models.** Compared with real-image-based method SupCon [50], it can be seen that there is a gap between training on real and synthetic *labeled* images (66.40 vs 57.16 Top-1 accuracy). This is probably due to the gap of image quality, as well as the lack of intra-class diversity in BigGAN synthetic images. Similar analysis has been provided in [78], where a classifier trained on BigGAN synthetic images gets similar results. This phenomenon is different from that (using synthetic images sampled from a high quality unconditional generative model for representation learning) in the main text, where the synthetic feature extractor can outperform the real feature extractor, indicting that there is still a long way to go when using pretrained *class-conditional* generative models for representation learning.

**Visualization Analysis.** In Figure 5, we give visualization comparison to analyze the behavior of our COP-Gen and other latent transformation methods. Firstly, the "Gaussian" and "Steering" method cause artifacts in a few images (e.g., row "park bench" column "Gaussian" and "Steering") due to some randomly picked aggressive transformations, which violets the input distribution of BigGAN. Next, let us compare our COP-Gen with the intuitively optimal "Random" method. The "Random" method looks well at first sight in most posted examples, since it draws independent samples as the positives. However, this strategy cannot identify and remove the nuisance information in each anchor. For example, in row "walking stick" and "little blue heron", "Random" method generates similar background with "Anchor", while our learning-based COP-Gen can address this problem as explained in the main text. Further, during the COP-Gen training process, it *learns to create* objects that is not exactly the anchor but relevant to the current class (e.g., the last two rows).These phenomena show that COP-Gen tries to use additional information to create more diverse positive pairs.

## F Computing Resources

Most of our experiments are conducted on 1~4 NVIDIA 2080 Ti GPUs. On ImageNet-1K, contrastive learning of 100 epochs takes ~3 days on 2 GPUs; linear evaluation on single GPU takes about 12 hours. Object detection on VOC takes ~1 day on 2 GPUs. Training COP-Gen in the latent space of pretrained BigBiGAN over $200K$ samples takes about 1 hour on 4 GPUs. Besides, sampling $1.3M$ image pairs offline (for latent-space-based approaches) takes about 6 GPU hours.

In addition, a few experiments that require large GPU memory are conducted on 4 NVIDIA 40GB A100 GPUs. These experiments include COP-Gen training in the latent space of pretrained Residual Flow and BigGAN. Other details are specified when introducing those experimental settings respectively.

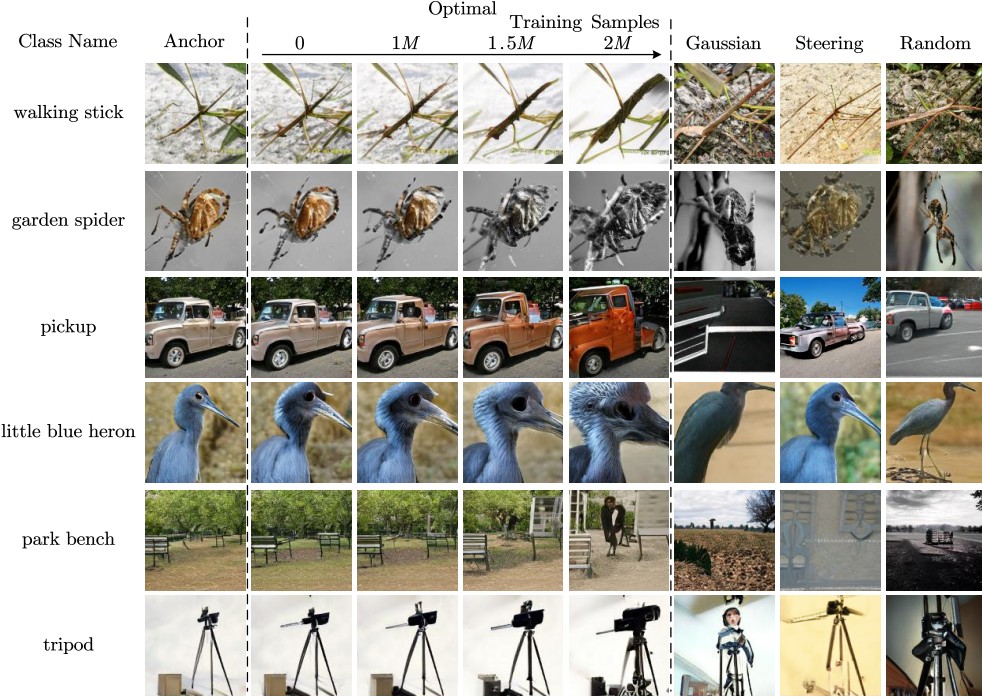

Figure 5: Left: Randomly sampled anchors from ImageNet-1K pretrained class-conditional BigGAN and their corresponding ImageNet labels. Middle: Generated anchor-specific positives by our COP-Gen approach **for supervised contrastive learning** with respect to the training steps. Right: Results by other latent transformation based positive generation methods.