# OpenReview forum: "Optimal Positive Generation via Latent Transformation for Contrastive Learning"
_NeurIPS.cc/2022/Conference — NeurIPS 2022 Accept_

### Official Review · Reviewer_yTbP · 2022-06-27

**Rating:** 5
**Confidence:** 3
**Soundness:** 3 good
**Presentation:** 3 good
**Contribution:** 3 good

**Summary:**

This paper proposes a new framework -  Contrastive Optimal Positives Generation (COP-Gen) for self-supervised contrastive learning which aims to learn instance-specific latent transformations that minimizes the mutual information between the generated positive pair subject to the semantic consistency constraint.

**Questions:**

See weaknesses above

**Limitations:**

adequately addressed

**Strengths And Weaknesses:**

### Strengths
1.  The paper is well written and easy to follow.

2. The idea of using a pre-trained generative network to obtain the positive samples is novel.

### Weaknesses
My only concern about this paper is in the experimental part.

1. I observed that the authors adopt a resolution of 112x112  images during the pertaining stage, and the reported performance for other methods is much lower than the numbers in the original papers. I think strictly following the setting from the original paper will be a fairer comparison. (e.g. 100 Epoch of SimCLR should reach 62.8% Top-1 Accuracy)

2. Does the quality of the generative model affect the final result?

3. What is the extra training cost if we include an additional generative model?

---

> ### Author Response · Authors · 2022-08-02
> **Author Response to Reviewer yTbP**
>
>
> Thank you for the review and acknowledgment of our clarity and novelty.  We address your questions and concerns below.
>
> **(1) Resolution of images.**
> - Since the released pretrained BigBiGAN [1] can only generate 128x128 images, comparing trained encoders on these synthetic images with 224x224 real image trained encoders will be unfair.  128x128 resolution is also adopted at linear evaluation stage to align with the pretraining stage, causing the relatively lower reproduced results.
>
> **(2) Does the quality of the generative model affect the final result?**
> - In theory, the final result can be improved with a perfect pretrained generative model. To simply verify this, we use IC-GAN [2], which achieves the state-of-the-art performance in IS and FID (widely used metrics to evaluate the similarity between generated and real images) on unconditional ImageNet generation, as the data source for contrastive learning.
> - The following tables show the results. Interestingly, for both GenRep baseline and our COP-Gen method, contrastive learning does not benefit much from the "better" pretrained generative model. This might be because of the mismatch in evaluation protocols between generative models' training and their usage in representation learning. We hope this could encourage the community to rethink the existing metrics of generative models. And how to design a prefect generative model for representation learning remains an open issue.
>
> **Contrastive learning on different generative models  (w/o using latent transformations):**
> | Method                | Pretraining Data | Resolution | IS $\uparrow$  | FID $\downarrow$ | ImageNet-100 Top-1 |
> | --------------------- | :--------------: | :--: | :------------: | :--------------: | :----------------: |
> | GenRep ($T_z$ = None) |     BigBiGAN     | 128  |   23.3 ± 0.2   |    28.5 ± 0.4    |     **58.26**      |
> | GenRep ($T_z$ = None) |      IC-GAN      | 128  | **48.7** ± 0.2 |  **11.7** ± 0.0  |     **58.26**      |
>
> **Contrastive Learning on different generative models  (using COP-Gen transformations):**
> | Method         | Pretraining Data | Resolution | IS $\uparrow$  | FID $\downarrow$ | ImageNet-100 Top-1 |
> | -------------- | :--------------: | :--: | :------------: | :--------------: | :----------------: |
> | COP-Gen (ours) |     BigBiGAN     | 128  |   23.3 ± 0.2   |    28.5 ± 0.4    |       63.96        |
> | COP-Gen (ours) |      IC-GAN      | 128  | **48.7** ± 0.2 |  **11.7** ± 0.0  |     **64.06**      |
>
>
> **(3) Extra training cost of generative models.**
> - Training generative models on large scale dataset like ImageNet is time consuming. For instance, training BigGAN for 128x128 images will take about two weeks on 8 Nvidia V100 GPUs [3]. Nevertheless, the community has released many well-trained generative models for researchers to use. And the main focus of this paper is leveraging well-trained generative models to perform contrastive learning. But it has to be admitted that this extra cost should be considered if we want to train a specific generative model for contrastive learning.
>
> **References**
> - [1] Jeff Donahue and Karen Simonyan. "Large scale adversarial representation learning." NeurIPS 2019.
> - [2] Arantxa Casanova, Marlene Careil, Jakob Verbeek, Michal Drozdzal, and Adriana Romero. "Instance-Conditioned GAN." NeurIPS 2021.
> - [3] Andrew Brock, Jeff Donahue, and Karen Simonyan. "Large scale GAN training for high fidelity natural image synthesis." ICLR 2019.

---

> > ### Comment · Reviewer_yTbP · 2022-08-09
> > **Thank you for the detailed response**
> >
> > Many thanks to the author's responses. My concerns have been addressed.

---

### Official Review · Reviewer_kg96 · 2022-07-11

**Rating:** 6
**Confidence:** 5
**Soundness:** 3 good
**Presentation:** 3 good
**Contribution:** 3 good

**Summary:**

This paper focuses on how to  generate Contrastive Optimal Positives (COP-Gen) for self-supervised contrastive learning and propose a new method based on pretrained generative models. Then, this paper presents theoretical analysis to illustrate that the proposed method can generate positives that removes as much nuisance information as possible while preserving the semantics. Experimental results show the effectiveness of the proposed method.

**Questions:**

1. A major problem faced by GANs is the mode collapse problem. This can be understood as learning only part of the semantic information. Therefore, using GAN for contrastive learning may lead to the problem of lack of semantics, so that it cannot better serve downstream tasks. How to solve this problem?
2. I am very confused about Lemma 3.3. If Lemma 3.3. holds, then the learned semantic features are related to the input $Z$. However, $Z$ is a random noise and contains no semantic information.
3. The caption of Figure 2 is puzzling.
4. I hope the authors can provide the code as soon as possible. I want to verify this method in imagenet-1K dataset.
5. How does the method perform on transfer tasks and semi-supervised learning tasks? I hope the author can give the corresponding experimental results.

**Limitations:**

The authors have adequately addressed the limitations and potential negative societal impact of their work.

**Strengths And Weaknesses:**

Strengths:
1. This paper is novelty, and the main idea is interesting.
2. The experimental results is good.

Weaknesses:
1. The technical contribution is incremental.
2. The theoretical analysis is vague. I think the learned transformations can not remove the task-independent semantic information only based on the Definition 2.1 and Proposition 3.1.
3. The compared contrastive learning methods are outdated.

---

> ### Author Response · Authors · 2022-08-02
> **Author Response to Reviewer kg96 (1/2)**
>
> Thank you for the review and insightful comments.  We are glad that you find the idea is interesting and novel. We will address your comments about the theoretical part first and then answer the other questions.
>
> **(1) The learned transformations can not remove the task-independent semantic information only based on the Definition 2.1 and Proposition 3.1.**
> - The optimal positives for contrastive pretraining do need to *remove downstream-task-independent information* (Definition 2.1). Unfortunately, the downstream tasks are usually unknown at pretraining stage. To this end, we turn to create positives that *removes semantic-irrelevant information of pretraining dataset at instance level*, hoping that to be helpful and general, if downstream tasks could benefit from the pretraining dataset. This was clarified in Footnote 2 of the paper.
> - Based on this, leveraging the remarkable property of unsupervised pretrained generative models (Proposition 3.1), we propose to remove as much shared information between the generated positives as possible while preserving the semantics (Eq. 5), whose optimality (under unknown-downstream-task setting) has been proved under some assumptions.
>
> **(2) Lemma 3.3 causes the learned semantic features are related to the input $z$. However $z$ is a random noise and contains no semantic information.**
> - $z$ is a random noise but the image $G(z)$ contains semantic information. And *the learned semantic feature at contrastive learning stage* **depends on what is shared between the positive pair $G(z)$ and $G(T_z(z))$**, instead of single $z$ or $G(z)$.
> - Lemma 3.3 ensures the invertibility of the pretrained generator $G$. Its role in the proof (*at COP-Gen navigator training stage*) can be understood as to ensure $G(T_z(z))$ can be mapped back to a deterministic place $T_z(z)$ in the latent space (see the diagram below). Thus training a navigator $T_z: z \to T_z(z)$ can make sense. Otherwise the navigator would be confused if there were more than one destination. We will make this clearer in the revised version.
>
> $\ {\small\text{data space}} \ \ \ \ \ G(z) \ \ \ \ \ \ G(T_z(z))$
>
> $\ \ \ \ \ \ \ \ \ \ \ \ \ \ \ \ \ \ \ \ \ \ \ \ \uparrow \ \ \ \ \ \ \ \ \ \ \ \uparrow$
>
> ${\small\text{latent space}} \ \ \ \ \ \ \ z  \ \ \ \to \ \ \ T_z(z)$
>
> **(3) Using GAN for contrastive learning may lead to the problem of lack of semantics, so that it can not better serve downstream tasks.**
> - An image generated by a pretrained GAN may actually contains less semantic information than a real image. As a consequence, *simply using the generated images* to perform contrastive learning will achieve a poor performance when compared with real-image-based method, as shown in the two first rows in Table 1 in the paper (SimCLR on real vs GenRep w/o $T_z$).
> - However, when GANs are introduced in contrastive learning, *the transformation in their latent spaces can be explored* to create more proper positive pairs, therefore the final performance (learned semantics) of contrastive learning depends on not only the image quality but also the creation of positive pairs, as emphasized in our answer to the previous question. The last three rows in Table 1 (GenRep Gaussian, "steering", and our COP-Gen) verified this, where latent transformations fill the performance gap caused by image quality. Besides, *these synthetic-image-trained encoders outperform real-image-trained encoders among multiple downstream tasks*, as shown in our response to your last question as well as Table 1.

---

> ### Author Response · Authors · 2022-08-02
> **Author Response to Reviewer kg96 (2/2)**
>
> **(4) Caption of Figure 2 is puzzling.**
> - Figure 2 is used to visually illustrate that the semantic of $G(z')$ will be different from $G(z)$ if $d(z, z')$ keeps raising. The latter part of the caption gives a realization of increasing the distance between $z$ and $z'$, i.e. sampling from Gaussians with different standard deviations, which is consistent with the baseline GenRep Gaussian.  We will move this part to the experiment section to make it clear.
>
> **(5) Provide the code.**
> - We provide our code and pretrained models in this [anonymous link](https://github.com/AnonResearcher0/Paper9126). Please refer to the readme file included and let us know if you have any questions on reproducing the results.
>
> **(6) Transfer learning and semi-supervised learning tasks.**
> - We conduct linear transfer classification experiment on several datasets. For transfer detection with full fine-tuning, please refer to Table 1 in our paper. For semi-supervised learning task, we follow the setting in SimCLR,  i.e., sample 1% or 10% of the labeled ImageNet training set in a class-balanced way, and fine-tune the whole network on the labeled data. The two new experiments are implemented based on the open-source code [VISSL](https://github.com/facebookresearch/vissl/) using its standard set of hyper-parameters and evaluation protocols.
> - We compare our method with the baseline SimCLR and two of the most competitive methods: NNCLR on the real and latent “steering” method.  The following two tables show the results.  It is worth noting that synthetic-image-trained encoders outperform real-image-trained encoders on nearly all transfer and semi-supervised benchmarks. And our method achieves competitive performances when compared with previous latent "steering" method.
>
> **Transfer classification results:**
> | Method          | Pretraining Data |   Food    |  CIFAR10  | CIFAR100  |  SUN397   |   Pets    | Caltech-101 |  Flowers  |
> | --------------- | :--------------: | :-------: | :-------: | :-------: | :-------: | :-------: | :---------: | :-------: |
> | SimCLR          |   ImageNet-1K    |   65.16   | **84.44** |   60.99   | **66.26** |   69.72   |    81.96    |   82.81   |
> | NNCLR           |   ImageNet-1K    |   62.53   |   84.10   |   60.43   |   64.49   |   70.35   |    82.45    |   82.84   |
> | GenRep Steering |     BigBiGAN     |   66.17   |   84.23   | **63.29** |   62.46   |   69.94   |  **84.65**  | **84.78** |
> | COP-Gen (ours)  |     BigBiGAN     | **67.05** |   84.09   |   62.44   |   63.82   | **73.56** |    84.63    |   84.36   |
>
>  **Semi-supervised learning results:**
> | Method          | Pretraining Data | Label fraction 1% Top-5 | Label fraction 10% Top-5 |
> | --------------- | :--------------: | :---------------------: | :----------------------: |
> | SimCLR          |   ImageNet-1K    |          45.77          |          77.34           |
> | NNCLR           |   ImageNet-1K    |          44.09          |          76.28           |
> | GenRep Steering |     BigBiGAN     |        **52.33**        |          78.70           |
> | COP-Gen (ours)  |     BigBiGAN     |          51.84          |        **78.81**         |

---

### Official Review · Reviewer_58Jb · 2022-07-12

**Rating:** 6
**Confidence:** 4
**Soundness:** 3 good
**Presentation:** 2 fair
**Contribution:** 3 good

**Summary:**

This work proposes an approach to generate optimal positives for contrastive learning. The positive pairs are generated by finding an instance-specific latent space navigator $T_z$ by minimizing the mutual information between the positive pairs.
The synthetic data thus generated is used for self-supervised contrastive learning where the proposed approach outperforms the prior work on the Imagenet dataset and on the downstream task of object detection on the Pascal-VOC dataset.

**Questions:**

1. The choice of a good $\delta$ in equation 5. The experiment section does not discuss the choice of $\delta$ used in the paper. How is it chosen and what impact does training with different $\delta$ have on the experiments in the paper. When increasing the number of samples in Figure 3 for training does a lower $\delta$ mean redundant samples? It would be good to analyze the performance of the approach of different $\delta$ and the different number of samples.

2. In line 39: "in contrast to representation space transformation and non-domain-knowledge-guided data augmentation"; what kind of domain knowledge would be better preserved in this case?

3. How the method compares to prior work in terms of computational efficiency (training time, resources) should be discussed in the main paper. Even though in the checklist, the method is claimed to be computationally expensive. The augmentation with the optimal positives would be useful to the community if it does not add too much computational overhead.

4. The experimental results in table 1 are well within the standard deviation of the prior work. While its understandable that the experiments are time-consuming, the effect of different random seeds should be studied to establish the usefulness and significance of the results.


**Limitations:**

The paper adequately discusses the limitations of the approach and its dependence on pre-trained models. The discussion on how the biases are amplified with the proposed augmentation approach can be discussed in further detail. For example, what happens when increasing the number of positive pairs in case of a biased dataset?

**Strengths And Weaknesses:**

+ The paper is easy to follow. The paper is well-written and motivated. The contributions of the work wrt to the prior work and finding instance-specific optimal positives for augmentation in self-supervised contrastive learning is clear.

+ The experimental results outperform prior work trained on synthetic images.  The experimental results in line with prior work, demonstrate the proposed approach outperforms the prior approaches. The experiments are also performed on VOC object detection to show the applicability of the approach in downstream tasks.

+ Ablations are thorough. The ablations conducted in Tables 2,3 and 4 show the benefits of the data augmentations as well as of the transformation applied in the latent space.

- There are no major weaknesses, apart from the missing discussions on the choice of certain parameters (see Questions below) and comparison to prior work in terms of computational overhead.

- The manuscript can be revised to fix the following typos:
   *Equation 1, function sim not defined.
   *Line 90: "behind by formulating" behind our approach?
   *Line 44: "computational" to computationally.
   *Line 239: "as did in [60, 15, 28]" to as done in.
   "Border Impact" to Broader Impact

---

> ### Author Response · Authors · 2022-08-02
> **Author Response to Reviewer 58Jb (2/2)**
>
> **(5) How the biases of datasets are amplified with the proposed approach?**
> - The discussion of the impact of contrastive learning and our approach on biased dataset is an interesting and meaningful topic. Take a class-imbalanced setting for example. Although vanilla contrastive learning can learn more balanced feature space than its supervised counterpart since it does not use labels [5], it still could not immune to imbalance. It is hypothesized that the high-frequency "classes" might dominate the learning process and leaves the low-frequency "classes" under-learned [6].
> - We here hypothesize that the learning of samples from low-frequency "classes" might benefit from COP-Gen generated positives. Because the training process of COP-Gen has seen the overall imbalanced distribution thus has the potential to take it into consideration to create more proper positives.
>
> **References**
> - [1] Ting Chen, Simon Kornblith, Mohammad Norouzi, and Geoffrey Hinton. "A simple framework for contrastive learning of visual representations." ICML 2020.
> - [2] Yonglong Tian, Chen Sun, Ben Poole, Dilip Krishnan, Cordelia Schmid, and Phillip Isola. "What makes for good views for contrastive learning?" NeurIPS 2020.
> - [3] Ali Jahanian, Xavier Puig, Yonglong Tian, and Phillip Isola. "Generative models as a data source for multiview representation learning." ICLR 2022.
> - [4] Ali Jahanian, Lucy Chai, and Phillip Isola. "On the “steerability” of generative adversarial networks." ICLR 2020.
> - [5] Bingyi Kang, Yu Li, Sa Xie, Zehuan Yuan, and Jiashi Feng. "Exploring balanced feature spaces for representation learning." ICLR 2021.
> - [6] Ziyu Jiang, Tianlong Chen, Bobak J Mortazavi, and Zhangyang Wang. "Self-damaging contrastive learning." ICML 2021.

---

> > ### Comment · Reviewer_58Jb · 2022-08-09
> > **Post Rebuttal Update**
> >
> > Thank you for addressing the comments. I have no further questions.

---

> ### Author Response · Authors · 2022-08-02
> **Author Response to Reviewer 58Jb (1/2)**
>
> Thank you for the detailed comments and constructive suggestions.  We are encouraged that you find the paper is well-motivated and experiments are thorough. We address your questions and concerns below.
>
> **(1) The choice of a good $\delta$ and its relationship with the number of training samples.**
> - Parameter $\delta$ is introduced to theoretically illustrate that the distance between $z$ and $z'$ should not be so large that the semantic changes significantly.
> - In practice, we do find that when increasing the number of training samples in Figure 3, $d(z, z')$ keeps increasing (Please check [these curves](https://github.com/AnonResearcher0/Paper9126/blob/master/figures/Tz_mean_std.png) which reflect the expectation of $d(z, z')$, in our response to question (3) of reviewer YrQG). Thus *setting a lower $\delta$ in theory can be understood as using smaller number of training samples or terminating the minimax training process earlier in practice*. The final performance w.r.t. the number of training samples in Figure 3 is provided in Table 4 of the paper correspondingly.
>
> **(2) What kind of domain knowledge would be better preserved for data augmentation?**
> - The best choice of data space augmentation used to create positive pairs is normally decided by large amount of experiments [1][2]. The domain knowledge here refers to the downstream task known in advance. For example, if the downstream task is to classify the color of objects, the choice of data augmentation should not contain color jittering, which will hurt the downstream performance.
>
> **(3) Discuss the computational overhead compared to prior works in the main paper.**
> - The detailed computational overhead of each stage of our method is listed in Appendix F.  We will discuss the extra computational overhead when comparing to prior works briefly in the main paper.
> - Overall, when comparing with non-learning-based latent transformation methods (GenRep Gaussian [3]), our method introduces additional latent navigator training overhead. This computational cost also happens in the previous learning-based latent "steering" method [3][4].
> - When comparing with real-image-based baseline [1], all the computations happen at the latent space of generative models are considered. This normally includes navigator training and positive (pretraining dataset) sampling. All these computations above are not too much (several GPU hours).  But if we take training generative models into consideration, the extra training cost would be expensive (typically several GPU weeks). Nevertheless, the community has released many well-trained generative models for researchers to use.
>
> **(4) Repeat the experiments in Table 1.**
> - Since contrastive pretraining experiments are time-consuming, we only rerun our method, the baseline SimCLR,  and the most competitive NNCLR in Table 1 for another two times (a total of three times).  For our method, the repetition is also conducted at the navigator training (COP-Gen) stage. The results with standard deviation are reported in the following table.  Although COP-Gen has a relatively higher standard deviation since the termination of navigator training is decided by simply monitoring the quality of generated positives, it achieves overall good performance against the most competitive real-image-based method NNCLR.
> | Method          | Pretraining Data | ImageNet-1K Top-1| ImageNet-1K Top-5|
> | --------------- | :--------------: | :--------------: | :--------------: |
> | SimCLR          |   ImageNet-1K    |   49.37 ± 0.07   |   75.58 ± 0.01   |
> | NNCLR           |   ImageNet-1K    |   52.18 ± 0.19   | **76.91** ± 0.11 |
> | COP-Gen (ours)  |     BigBiGAN     | **52.64** ± 0.74 |   76.59 ± 0.63   |

---

### Official Review · Reviewer_YrQG · 2022-07-13

**Rating:** 5
**Confidence:** 3
**Soundness:** 2 fair
**Presentation:** 3 good
**Contribution:** 2 fair

**Summary:**

The work follows the GenRep paper and performs representation learning based on a generative model trained on data. They extends the GenRep by switching the fixed latent transformation to a learnable transform. They formulates COP-Gen as an instance-specific latent space navigator which minimizes the mutual information between the generated positive pair, and trains the encoder by maximizing the mutual information between positive pairs. It is clearly written and the theoretical part is correct. The idea is easy to follow and widely used in other frameworks such as GAN model. The performance is improved compared to the GenRep paper, but to be expected.

**Questions:**

- Why do you choose 224 as batch size? It might be better to align with other people's work and use 256 instead.

- Could we combine this hard positive technique with non-contrastive learning frameworks, such as BYOL and SimSiam? Are there any experiment results on a small dataset, if the time is limited?

- Why not use a Lagrange multiplier to contain the semantic distance between z and z'?

- Normally training the model in a minimax way takes a longer time (epochs) to converge, or even struggles to converge. Could you provide a training/test loss plot with respect to the epochs?

**Limitations:**

The authors have adequately addressed the limitations and potential negative social impact of their work.

**Strengths And Weaknesses:**

Strengths:
- In previous works, people tries to improve the performance of contrastive learning by creating hard negative, while in this paper, the authors uses the InfoMin criterion to create optimal positives. The authors provided solid mathematical results to support the idea, and the proposed method is described and illustrated clearly.

Weaknesses:
- The performance is highly limited by the trained BigBiGAN model, although in theory with a perfect GAN model we could expect a comparable or even better results compared to SimCLR paper.

- Minimax training objectives might struggles to converge to the optimal point, but this paper did not touch on this point. It would be helpful if more details about the training process is provided.

- Some non-contrastive paper, such as BYOL and SimSiam should be included in references.

- Minor point:
  Typo: Border Impact ---> Broader Impact

---

> ### Author Response · Authors · 2022-08-02
> **Author Response to Reviewer YrQG (2/2)**
>
>
> **(4) Use a Lagrange multiplier to constrain the semantic distance between $z$ and $z'$.**
> - Methodologically, we can use a Lagrange multiplier to convert the semantic constraint $d(z, z') \leq \delta$ into a regularization term, and the training objective becomes $\min_{T_z} I(t_x(G(z)); t'_x(G(z'))) + \lambda d(z, z')$, where $T_z(z) = z' - z$ in practice.  Note that this introduces an additional hyper-parameter $\lambda$ which might be difficult to tune.
> - If $d(z, z')$ dominates the training process ($\lambda$ is large), it will make $z'$ so close to $z$ that the navigator $T_z$ could not remove enough nuisance information thus the optimal could not be achieved.
> - If $I(t_x(G(z)); t'_x(G(z')))$ dominates ($\lambda$ is a small or reasonable value), as shown in the curve of the previous answer, it will make $d(z,z')$ gradually increase. The training process might struggle to converge and the objective might degrade to the original form.
>
> **References**
> - [1] Jean-Bastien Grill, Florian Strub, Florent Altché, Corentin Tallec, Pierre Richemond, Elena Buchatskaya, Carl Doersch, Bernardo Avila Pires, Zhaohan Guo, Mohammad Gheshlaghi Azar, Bilal Piot, koray kavukcuoglu, Remi Munos, and Michal Valko. "Bootstrap your own latent: A new approach to self-supervised learning." NeurIPS 2020.
> - [2] Yonglong Tian, Chen Sun, Ben Poole, Dilip Krishnan, Cordelia Schmid, and Phillip Isola. "What makes for good views for contrastive learning?" NeurIPS 2020.
> - [3] Ali Jahanian, Xavier Puig, Yonglong Tian, and Phillip Isola. "Generative models as a data source for multiview representation learning." ICLR 2022.

---

> ### Author Response · Authors · 2022-08-02
> **Author Response to Reviewer YrQG (1/2)**
>
> Thank you for the review and helpful suggestions.  We are glad that you find the idea is easy to follow and the theoretical part is solid. We address your questions and concerns below.
>
> **(1) 224 as batch size.**
> - Our setting of the batch size is limited by the GPU memory. Most of our contrastive pretraining experiments are run on two 11GB 2080 Ti GPUs. 224 is the maximum batch size that can be achieved. We will get more GPUs and conduct experiments on larger batch sizes, e.g., 256, to make the comparison more complete.
>
> **(2) Combine with non-contrastive learning frameworks.**
> - Combining our *contrastive* optimal positive generation method with *non-contrastive* frameworks is an interesting attempt. We conduct experiments on ImageNet-100-scale BigBiGAN synthetic images and evaluate linear performance on ImageNet-100 (setting in Section 4.3). We change the original framework to BYOL [1], based on the open source codebase [Lightly](https://github.com/lightly-ai/lightly). The following table shows the results. It can be seen that changing the self-supervised learning (SSL) framework has little impact to pretraining on synthetic dataset. When combining with non-contrastive BYOL, our COP-Gen could still improve the learned representations, which empirically indicates that *the optimal positive pairs for non-contrastive approaches might share the same InfoMin principle [2] with contrastive approaches*.
> | Method         | SSL Framework |  $T_z$  |    $T_x$    | ImageNet-100 Top-1 |
> | -------------- | :-----------: | :-----: | :---------: | :----------------: |
> | GenRep         |    SimCLR     |  None   | SimCLR Aug. |       58.26        |
> | COP-Gen (ours) |    SimCLR     | Optimal | SimCLR Aug. |     **63.96**      |
> | GenRep         |     BYOL      |  None   | SimCLR Aug. |       56.34        |
> | COP-Gen (ours) |     BYOL      | Optimal | SimCLR Aug. |     **64.46**      |
>
> **(3) Details about the minimax training process.**
> - We provide the plot of minimax training loss in this [anonymous link](https://github.com/AnonResearcher0/Paper9126/blob/master/figures/minimax_loss.png), where the mutual information estimator ($f$) tries to minimize the InfoNCE loss, while the latent navigator ($T_z$) tries to maximize that. It can be seen that the two losses act as mirror images since they are competing against each other. It's ok for the losses to shake a little bit, which is the evidence of the model trying to improve itself.
> - As the training goes, some useful information can be observed from the plot. At the beginning (0-170k training samples),  the loss of $f$ keeps decreasing, which means that the mutual information learned by $f$ exceeds the mutual information drop caused by $T_z$ learning, and $f$ has not been well-trained during this period. In the middle part (170k-800k), the loss of $f$ / $T_z$  keeps increasing / decreasing, which means that $T_z$ dominates this learning period, and $f$ might have been well-trained. At last (800k-the end), the two losses gradually converge. Only from the minimax curves, it is hard for us to tell when $T_z$ has been trained to be optimal. All we know is that we should pick $T_z$ after training over 170k samples, since the mutual information estimator has not been well-trained.
> - A simple and effective way to solve this is to check the generated positive pairs, as shown in Figure 3 of the paper. We also monitor the mean ($\mu$) and standard deviation ($\sigma$) of $T_z(z)$ (in practice $z' = z + T_z(z)$) of each batch during training. This is because the expectation of $d(z, z')$ can be estimated as $\mu^2 + \sigma^2$, as discussed in Section 3.3 of the paper. As shown in [these curves](https://github.com/AnonResearcher0/Paper9126/blob/master/figures/Tz_mean_std.png), the mean keeps around zero while the standard deviation keeps raising, indicating that $d(z, z')$ keeps raising during the training process.  When the value is too large compared with the prior value investigated in [3], the training process should be terminated. We use this as an auxiliary monitor signal. These practices are summarized in Appendix B and we will analyze more with the training curves in the revised version.

---

### Author Response · Authors · 2022-08-02
**Summary of the Responses to All Reviewers**

We thank all reviewers for the time and effort in reviewing our paper and the insightful feedback. We are encouraged that reviewers find our paper is well-motivated and easy to follow, and the idea is interesting and novel. We have faithfully made detailed responses to the reviewers, provided our code and models, and conducted additional experiments:

 - Combine our method with non-contrastive learning framework
 - Repeat the experiments and report error bars
 - Evaluate on transfer learning and semi-supervised learning tasks
 - Investigate the performance affected by pretrained generative models

We thank you for these helpful suggestions and we find these experimental settings and results are meaningful and valuable. These discussions further strengthen our paper and will be added in the revised version.

---

### Meta-Review · Area_Chair_siMb · 2022-08-30

**Recommendation:** Accept
**Confidence:** Less certain

**Metareview:**

This is a theory-oriented paper for contrastive representation learning. It extends the GenRep by replacing the fixed latent transformation to a learnable transform. The results on ImageNet-1K image classification linear probing and VOC object detection demonstrate the effectiveness of the proposed methods.

The paper receives unanimous accept from all reviewers, leading to an ``Accept'' decision. However, the impact of this paper could be higher (by attracting more attention), if it can compare with absolute SoTA methods in the leaderboard [*] in computer vision community.


[*] https://paperswithcode.com/sota/self-supervised-image-classification-on

**Award:**

No

---

### Decision · Program_Chairs · 2022-09-14

Accept